# Systematic Observation of Emotional Regulation in the School Classroom: A Contribution to the Mental Health of New Generations

**DOI:** 10.3390/ijerph20085595

**Published:** 2023-04-20

**Authors:** Marina Alarcón-Espinoza, Paula Samper-Garcia, M. Teresa Anguera

**Affiliations:** 1Psychology Department, Universidad de La Frontera, Temuco 4780000, Chile; 2Basic Psychology Department, Universitat de València, 46010 Valencia, Spain; paula.samper@uv.es; 3Faculty of Psychology, University of Barcelona, 08028 Barcelona, Spain; mtanguera@gmail.com

**Keywords:** emotional education, preadolescents, school coexistence, mixed methods, citizenship

## Abstract

Emotional regulation is a developmental milestone, as it promotes well-being throughout life. Children between 10 and 12 years old are expected to reach capacities that allow them to regulate themselves emotionally, the school context being a natural setting for this challenge. With the objective of analyzing the forms of expression and regulation of emotions that are observed in the school classroom, this research was conducted through a *mixed methods* study that systematically observed nine classes during five sessions each. The design was Nomothetic, Follow-up and Multidimensional; the observations were recorded on audio and in person and were later transformed into data by coding them according to an ad hoc designed instrument. The concordance of the records was evaluated, a sequential analysis of delays (GSEQ5) was performed to detect regularities and existing sequences, and a polar coordinate analysis (HOISAN) observed the relationships between the categories. Finally, the presence of multiple cases was detected. The results detail the ways in which different actors express emotions and interact, regulating the emotions of other people. The results are discussed based on the need to foster educational intentionality and allow students’ emotional self-regulation.

## 1. Introduction

In order to develop emotionally, children need to learn how to regulate their level of emotional arousal, learn how to balance individual concerns and more socially oriented concerns, and how to express emotions adaptively. In the learning of these skills, it is the social environment that teaches the child when to feel and how to express it [1], and a fundamental milestone in this process is to achieve emotional regulation [2,3]. Developing these emotional self-regulation skills is a complex capacity that is neither good nor bad but adaptive, depending on the demands of the context and the individual or social purposes pursued [4], and it fosters violence-free social and couple relationships [5].

Ordóñez and González [6] state that, in order to achieve such regulation, it is necessary to initially advance in the development of emotional awareness, which refers to becoming aware of one’s own emotions and the emotions of others, is the most basic emotional competence, prior to the development of other competencies. In this way, emotional awareness allows the interruption of the emotional program in progress, gaining flexibility and adapting to the extent where he or she can choose the most appropriate response to each situation [7].

In this sense, much of the progress in the development of emotional awareness and regulation is related to the fundamental evolutionary milestone of school age when attending school, which involves integration into the peer group and fosters socio-emotional interactions, the internalization of social norms, and social comparison processes [7]. Attending to the emotions of others is one of the most culturally sensitive components, which is very relevant to consider when observing that many social, emotional, and behavioral problems are related to social, cultural, and educational norms and attitudes [8].

It has been observed that the social support on offer within the school affects the students’ perception of subjective well-being, underscoring the importance of accepting and being aware of emotional responses in the construction of social relationships in the school, which involves strategies that are linked to feelings of being accepted, respected, included and supported by others [9].

When students have more adult support in school, they exhibit more positive emotions during learning activities as well as greater effort, focus, and persistence by initiating and taking part in learning activities. In this regard, Frivold Kostøl and Cameron [10] noted that in order to mitigate the possible adverse effects of emotion, teachers could develop behaviors that support effective emotional regulation: (a) help students control their behavior when they are angry, (b) facilitate emotional awareness in students and promote an acceptance of emotion through participation in conversations and debates with students on a range of emotional responses in the classroom and life events, and (c) allow the expression of emotions that the student can experience, supporting them to express their thoughts and feelings.

According to Riquelme [11] and Riquelme et al. [12], this can be conducted through strategies such as theater, music, or critical reading; or favor peer dialogue processes as a way to internalize important knowledge [13].

Thus, by regulating themselves emotionally, people achieve greater social integration, which contributes to the possibility of obtaining greater social support in the face of possible difficulties, favoring their mental health [14,15].

Consequently, Bisquerra et al. [16] emphasized the importance that the processes of training and emotional development include values and ethical principles since having good emotional competencies does not guarantee that they will be used for good and not evil, emphasizing the role of emotional competencies in active, effective, and responsible citizenship.

This implies reviewing not only the processes of self-regulation in children but also the relationships they establish with adults and the modeling that adults exhibit when relating to each other in everyday life within contexts such as school [17,18,19]. In this regard, Bisquerra [20], y Burguet, and Buxarrais [17] emphasized the role of the teachers, considering them as essential in the dimension of emotional support that can be provided in the learning process in order to foster the full development of an individual’s integral personality and form learning based on communicative guidelines that promote interaction, cooperation, and respect for others. Such learning involves the development and well-being of individuals and their communities in a constructive and meaningful way [21,22,23,24].

In this regard, it is necessary to mention that, according to García-Lázaro et al. [25], schools have not considered emotional education, focusing on conveying theoretical knowledge and attending to emotional issues basically when dealing with affective and/or behavioral problems in the diversity of the school classroom. Therefore, the importance of training all teachers in the observation and understanding of their affective world in their role as educators should be emphasized since this learning would be transferable to the daily interactions of the educational center, stressing that emotional education cannot be a matter of some specialized personnel, and, hence, must be present in all educational relationships that are established [26].

Thus, assuming that emotional development and the achievement of self-regulation in children are modeled and influenced by the contexts in which their daily lives are developed, where school must have great influence, the objective of this research is to analyze the forms of expression and regulation of emotions observed in the school classroom.

## 2. Material and Method

The observational methodology was chosen as the scientific method, using the indirect observation modality [27]. This methodology, considered a *mixed methods* in itself [28,29], seeks to integrate qualitative and quantitative elements using the *quantizing* proposal of Anguera et al. [30]. This approach can transform the information differently to how mixed methods are described in the literature and consist of the succession of three stages QUAL-QUAN-QUAL. In the first QUAL stage, descriptive records are obtained, and from both this and the theoretical framework, a defined structure—an observation instrument—is built to systematize these in the form of a code matrix. In the QUAN stage, the code matrices are analyzed quantitatively, and in the second QUAL stage, the results are interpreted, returning to the initial problem.

### 2.1. Design

The observational design [31] used was Nomothetic: nine course groups (cases) were observed. Follow-up: each course group was observed during five consecutive weeks in one hour of class (intersessional follow-up), and each observation was conducted during the entire class (intrasessional monitoring). Multidimensional: four major dimensions were observed by means of the observation instrument.

### 2.2. Participants

The groups observed corresponded to three educational establishments in the region of La Araucanía, Chile, as detailed in Table 1.

### 2.3. Procedure

Once authorization and informed consent had been obtained from the principals, teachers, parents, and students of each establishment, each class was observed with two instructors. The classes were audio-recorded and then transcribed into an Excel format in order to be initially segmented and later coded.

The segmentation was conducted considering the interlocutory (speaker) and syntactic (syntagma) criteria [28], giving priority to the speaker.

The coding was performed using the ad hoc designed instrument that combines field format with category systems [32], the structure of which was prepared from the descriptive records and the theoretical framework. It gradually generated conceptual macro-dimensions deployed in dimensions and subdimensions, and from each of which a category system (exhaustive and mutually exclusive) or behavior catalog (mutually exclusive) was constructed. This observation instrument called the “Guideline for the Observation of Communication and Emotional Self-regulation” (OCAE), was formed by the four following macro-dimensions: (A) knowledge of one’s own emotions and those of others, (B) self-regulation of behavior for the objective to be achieved, (C) the ability to establish assertive communication, and (D) an effective approach to conflict. Each of these was deployed at several levels, and the respective category systems and behavior catalogs were prepared until progressively obtaining the definitive observation instrument [33].

#### Reliability Analysis

In order to guarantee the quality of the data obtained and considering that in the scientific literature in different areas, it is common to use a percentage of records included between 5 and 20% to find an agreement (Lapresa et al. [34], 15% of the data were recorded. To guarantee the greatest rigor in the quality control of the data, the records were coded at three different times, and Cohen’s kappa coefficient was calculated [35], obtained by a computer using the free program GSEQ5, with multi-event data, type II [36], and achieving a concordance of 0.60 or greater.

### 2.4. Data Analysis

After assuring the quality of the data obtained, two quantitative analysis techniques were applied: lag sequential analysis and polar coordinate analysis.

In order to respond to the research objective, the analyses observed the significant correlations between the behavioral criterion of dimension A of the OCAE instrument, referring to the knowledge of one’s own and other people’s emotions, and conditioned behaviors indicated in the other categories of the instrument. Finally, the observation of multiple cases was conducted.

#### 2.4.1. Lag Sequential Analysis

The lag sequential analysis was conducted by searching for sequential contingencies between categories or codes of behavior, which were recorded according to the order parameter, and previously coded on the basis of an observation instrument. Its objective is the detection of sequential patterns of behavior [37]. In this way, using the free program GSEQ, v. 5.1.23 [38] (https://www.mangold-international.com/en/products/software/gseq.html (accessed on 19 April 2023)), it is possible to detect regularities in the behavior from a statistical contrast between the conditional probabilities and expected probabilities, from the proposal of one or several given behaviors. In this way, the calculation of adjusted residuals between the given behavior and the conditioned behaviors was conducted, called lag 0 (R0), in which only the occurrences of the given behaviors were studied, and retrospective lags (R − 1, R − 2, …) and prospective lags (R + 1, R + 2, …) were those that would probably be associated with such given behavior, both retrospectively and prospectively.

Likewise, in each of the lags, activating and inhibiting conditioned behaviors around the behavioral criterion could be obtained, each requiring to be considered as such that their level of significance was greater than 1.96 (≥1.96) in the case of activating behaviors, and less than minus 1.96 (≤1.96) in the case of inhibiting behaviors.

#### 2.4.2. Polar Coordinate Analysis

The polar coordinate analysis is a data reduction technique proposed by Sackett [32] that starts from the results of fitted residuals obtained in the lag sequential analysis. The focal behaviors (code/category to be found in the center of the map of interrelationships to be obtained) are those that have been found to be significant in the lag sequential analysis, and the other categories of the instrument are considered conditioned behaviors.

For the previously mentioned, as proposed by Sackett [39], the Zsum parameter proposed by Cochran [40] was applied to the Z values obtained from the prospective and retrospective adjusted residuals to quantify the intensity of the associative relationship between the focal behavior and each of the conditioned behaviors, both prospectively and retrospectively. These Zsum values allowed us to vectorize the behavior by calculating the length and angle of the respective vectors (conditioned behaviors).

To conduct the polar coordinate analysis, a free program HOISAN, v. 1.6.3.3.4 was used [41], which provided the values of the prospective and retrospective Zsum parameters for each conditioned behavior, as well as the length and angle values of each vector and the quadrant in which it was located.

The vectors were interpreted in terms of their length and angle. Regarding the length, they were considered significant when their value was greater than 1.96 (for a significance level of 0.05), and the angle of the vector indicated the nature of the relation it established with the focal behavior, according to the quadrant in which it was located, with the following associative relations being observed: (a) Quadrant I: Focal behavior and conditioned behavior activate each other, both prospectively and retrospectively; (b) Quadrant II: Focal behavior inhibits conditioned behavior, and conditioned behavior activates focal behavior; (c) Quadrant III: Focal behavior and conditioned behavior inhibit each other, both prospectively and retrospectively; (d) Quadrant IV: Focal behavior activates conditioned behavior, and conditioned behavior inhibits focal behavior.

#### 2.4.3. Multi-Case Analysis

Once both analyses were conducted for each of the classes, the coincidences of the conditioned behaviors for each of the criterion/focal behaviors were reviewed, highlighting those that occurred on three or more occasions and which allowed the presence of multiple cases to be detected [42,43,44].

## 3. Results

Initially, the results observed in the lag sequential analysis were presented, which synthesized multiple cases for each category and were considered as given behavior (in bold). Subsequently, the results of the polar coordinate analysis were shown, considering the focal behaviors of the clear (A11) and confused (A12) expression of emotions, the actions of associating personal emotions with the behaviors of oneself (A21), showing awareness of other people’s emotional reactions (A31), relating other people’s emotions with their behaviors (A41) and differentiating between personal emotions and those of other people (A51).

### 3.1. Lag Sequential Analysis

A clear expression of emotions (A11) is usually a behavior performed by a female student (E3), as noted in Table 2.

The confused expression of emotions (A12) is a behavior performed by several students expressing the same idea at the same time (E5) and/or a student (E4), inhibiting the participation of the head teacher (E1). Before the confused expression of emotions (A12) occurs, in R − 1, the head teacher (E1) participates, inhibiting the participation of several students expressing the same idea at the same time (E5); in R − 2, the participation of the head teacher (E1) is inhibited. After the confused expression of emotions (A12), in R + 1, the head teacher (E1) speaks, inhibiting the participation of a female student (E3) and/or of several students expressing the same idea at the same time (E5); in R + 2, the head teacher (E1) is inhibited; in R + 3, the head teacher (E1) speaks while the participation of a female student (E3) is inhibited; and in R + 4, the head teacher (E1) is inhibited again.

Expressing emotions in a confusing way (A12) is usually conducted by several students expressing the same idea at the same time (E5) or by one student (E4), while the participation of the head teacher (E1) is inhibited. Before emotions are expressed in a confusing way (A12), in R − 1, the head teacher (E1) participates, inhibiting the participation of several students expressing the same idea (E5) or of a female student (E3); and in R −2, the participation of the head teacher (E1) is inhibited. After emotions are expressed in a confusing way (A12), in R + 1, the head teacher (E1) speaks, inhibiting the participation of a female student (E3) and/or several students expressing the same idea at the same time (E5); in R + 2, the participation of the head teacher (E1) is inhibited; in R + 3, the participation of a female student (E3) is inhibited; and in R + 4, the participation of the head teacher (E1) is inhibited again. The results associated with the subdimension confused expression of emotions (A12) are detailed in Table 3.

Regarding the association between personal behaviors with one’s emotions (A21), it can be observed that after someone performs this behavior (A21), murmurs (S1) and appears in R + 2, which is summarized in Table 4.

Showing awareness of other people’s emotional reactions (A31) is an action developed mainly by the head teacher (E1), inhibiting the participation of a student (E4). Previously, in R − 2, the head teacher (E1) participated. After showing awareness of other people’s emotional reactions (A31), in R + 1, a student would participate (E4), and/or a pause of the silence of more than one second would occur (C2142), inhibiting the possibility of the head teacher’s participation (E1); in R + 2 the head teacher (E1) would participate; in R + 3 a student (E4) would participate; and in R + 4 the head teacher (E1) would speak, as shown in Table 5.

Relating other people’s behaviors with their emotions (A41) is an action performed by the head teacher (E1) and/or the assistant teacher (E2), inhibiting the participation of several students expressing the same idea at the same time (E5). Previously, in R − 1, a female student (E3) participated, inhibiting the participation of the head teacher (E1); and in R − 2, the head teacher (E1) would speak. Subsequently, in R + 1 and R + 3, a female student (E3) would speak. This is summarized in Table 6.

The behavior of differentiating between personal emotions and those of other people (A51) appeared with significant values in five of the nine classes, and three or more similarities were not observed among the observed classes when reviewing the lags obtained.

### 3.2. Polar Coordinate Analysis

In the polar coordinate analysis, it was observed that the clear expression of emotions (A11) (Table 7) was mutually activated with the actions of showing awareness of other people’s affective reactions (A31), with the use of expressive language (C12) and the action of relating other people’s behaviors to their emotions (A41).

This clear expression of emotions (A11) was mutually inhibited by the use of informative language (C11) and the presence of parallel conversations (S2).

Along with this, it was possible to observe that the clear expression of emotions activated the action of proposing solutions when addressing a problem (D21) and that, when addressing a problem by proposing a solution (D21), a clear expression of emotions (A11) was inhibited.

The confused expression of emotions (A12) (Table 8) is mutually activated with the use of expressive language (C12), with the actions of relating other people’s behaviors to their emotions (A41), identifying a problem (D1), addressing the problem by showing empathy (D35), addressing a problem without proposing a solution (D24), showing awareness of the affective reactions of others (A31), addressing a problem by proposing a solution (D21), showing calmness when addressing a problem (D31), the use of directive language (C13), the possibilities of redundancy in the topic of conversation (B13), of favoring or calling on students to self-regulate their participation in the classroom (B23), of regulating the participation of specific individuals (B24) and addressing a problem by showing resentment (D32), inhibition (D33) or aggressiveness (D34).

The confused expression of emotions (A12) is mutually inhibited by the use of informative language (C11), the presence of murmuring (S1), and parallel conversations (S2).

Associating personal behaviors with one’s own moods (A21) (Table 9) is mutually activated with the action of relating other people’s behaviors to their affective states (A41), with the use of informative language (C11), the behavior of proposing solutions when addressing a problem (D21), as well as the behavior of differentiating between personal emotions and those of other people (A51), identifying a problem (D1), contributing to the topic of conversation (B11), and addressing a problem by showing empathy (D35).

Associating personal behaviors with an individual’s own mood states (A21) is mutually inhibited by the presence of parallel conversations (S2) and murmuring (S1).

Showing awareness of other people’s affective reactions (A31) (Table 10) is mutually activated with the behavior of relating other people’s actions to their emotions (A41), with both the clear expression of emotions (A11) and confused expression of emotions (A12), the action of contributing to the topic of conversation (B11), the use of both informative language (C11) and expressive language (C12), as well as showing awareness of the affective reactions of others. This is mutually activated by the behaviors of identifying a problem (D1), addressing it without proposing solutions (D24) and/or addressing it by showing empathy (D35).

Showing awareness of other people’s emotional reactions (A31) is mutually inhibited by the presence of parallel conversations (S2), the use of directive language (C13), the participation of the assistant teacher (E2), as well as the behaviors of changing the topic of conversation (B12), and/or encouraging students to self-regulate their participation in the classroom (B23).

Relating other people’s behaviors to their emotional states (A41) (Table 11) is mutually activated with the confused expression of emotions (A12) and the action of showing awareness of other people’s emotional reactions (A31); in addition, it is mutually activated with the participation of a female student (E3), the clear expression of emotions (A11), the association between personal emotions and one’s own behaviors (A21), differentiating between personal emotions and those of others (A51), contributing to the topic of conversation (B11), using expressive language (C12), identifying a problem (D1), and approaching a problem with empathy (D35). It can also be mutually activated with the use of informative language (C11) and/or providing a solution to a problem (D21) and the participation of a student (E4).

Relating other people’s emotions to their behaviors (A41) is mutually inhibited with parallel conversations (S2), with directive language (C13), the behavior of favoring students’ self-regulation in their participation in the classroom (B23), and/or with the action of proposing a solution when addressing a problem (D21).

The behavior of differentiating between personal emotions and other people’s emotions (A51) (Table 12) can be mutually activated with the behavior of relating other people’s emotions to an individual’s own behavior (A41) and associating personal emotions to one’s own behavior (A21). Making a differentiation between personal behaviors and other people’s behaviors (A51) is mutually inhibited by the presence of parallel conversations (S2).

## 4. Discussion

Regarding the forms of expression and the emotions of different actors within the classroom, the presence of differentiated styles of emotional expression is noteworthy. Thus, it can be seen that a female student (E3) probably shows emotions clearly (A11), that a male student (E4) expresses emotions confusedly (A12), that when several students express the same idea at the same time (E5), they may express emotions confusedly (A12), using an expressive language (C12); and that the head teacher probably seeks to inhibit the confused expression of emotions (A12), showing an awareness of others’ affective reactions (A31) and/or relating other people’s behaviors to their emotions (A41), through the use of directive language (C13). In this regard, it can be considered that these differentiated styles for each of the actors present in the classroom show the way in which they emotionally regulate themselves and the group by means of the type of language used and/or the clarity or confusion with which they express themselves emotionally, observing the presence of the head teacher as a main regulating figure. In this way, the reality is being built where emotional expression seems to require control, thus, not constituting an opportunity to promote learning that fosters interaction, respect, and cooperation, as has been proposed by [21,22,23,24].

In this sense, it is possible to observe orderly relationships between behaviors that, when presented in succession, show how emotions are regulated within the classroom. This clear expression of emotions (A11) with any type of language (informative, expressive, or directive) inhibits the presence of murmuring (S1) or parallel conversations (S2); the confused expression of emotions (A12), although inhibited by the participation of the head teacher (E1), also manages to inhibit the individual’s participation since it inhibits the possibilities of talking about “what happens to us”, inhibiting the possibility of relating other people’s emotions to their behaviors (A41): an action mainly performed by the head teacher. In this regard, it can be considered that emotional expression has a certain impact when observed in the classroom, perhaps because it is generally presented as a surprising element that requires attention both to regulate the behaviors and emotions of the group and, at the same time, to negotiate the subsequent actions and relationships that need to be established [45].

In connection with the above, Martínez-Maldonado et al. [46] noted a strong weight of the school context and student characteristics in the teaching practices observed and in the classroom interactions, which primarily present the high direction of the teacher towards the students, an absence of student–student interactions and a high student–knowledge interaction time. In this aspect, and considering the contributions of Marchesi [8] and Villanueva and Górriz [7], it was estimated that the presence of emotional expression in the classroom could be an opportunity to favor the development of emotional awareness to the extent that a reflective process was performed, with an expressive and loving language that allows the expression of all participants and seeks the achievement of the common good [16].

Meanwhile, what can be seen is that the emotions that are present in the various interactions within the classroom are not accompanied, most of the time, by reflective processes that allow for self-regulation, but, in some way, there was an attempt to inhibit them by establishing a certain dispute or counterpoint with the use of directive language and the action of talking about other people’s emotions and behaviors by the head teacher. This pedagogical practice, constituted by objectives and procedures oriented by a set of values [47], seemed to convey a marked valuation of tranquility in the classroom and school discipline, and a prioritization of content delivery was probably observed.

In this sense, observing a large amount of energy invested both in emotional expression and in the use of directive language to regulate it, it was worth wondering about the need to stop and reflect further on the teaching-learning process and the design of learning experiences when adapted to the needs of students [48]. In this regard, it was observed that the presence of more emotionally reflective behaviors, such as the association between personal emotions and one’s own behavior (A21) and the differentiation between personal emotions and those of other people (A51), were rarely observed in the school classroom. However, after conducting an analysis of polar coordinates, it was noteworthy that when these behaviors occurred, along with favoring the relationship between other people’s emotions and their behaviors (A41), they inhibited the presence of murmuring (S1) and/or parallel conversations (S2), favoring a more reflective process, especially when proposing solutions to any possible problem (D21). Likewise, in this aspect, it seems that the course group itself seeks to regulate emotional expression through the use of parallel conversations (S2) since it was observed that all the behaviors/categories of dimension A of the OCAE instrument (referring to emotional expression and regulation) were mutually inhibited by the parallel conversations (S2).

## 5. Conclusions

This research, along with showing the relevant presence of emotions in the classroom, exposes the need to regulate them, highlighting the statements of Gómez and Calleja [4] regarding the need for children to have life experiences that allow them to gradually internalize regulatory experiences, for teachers to have the necessary psychological well-being to regulate students’ emotions [49], and the important role of the school as a socializing context for the formation of values, as indicated by Garaigordóbil [50].

In this sense, the suggestions by Ferreira et al. [51] become relevant insofar as socioemotional learning must be based on a systemic perspective, where the development of socioemotional skills occurs not only in the classroom but also in the entire school, in families, and in the school community. This fully contributes to the education process since, as Berg et al. [52] and Greenberg et al. [53] indicate, the relation between school climate and socioemotional competencies is bidirectional, i.e., in the same way, that the school climate affects or influences socioemotional competences and socio-emotional competences affect or influence the school’s climate.

## 6. Limitations

Within the limitations of this research, it was considered that it has only been possible to record the information in audio because, in order to protect the identity of the participants, it has not been possible to have access to films that could have allowed the detection of gestures and other details, which could be possible in future research.

## Figures and Tables

**Table 1 ijerph-20-05595-t001:** Distribution of observed groups.

Location	School Funding	Grade	Number of Students	Subject
Fishermen’s Cove (2000 inhabitants)	Public	5º	15–16	Class Council (Tutorships)
		6º	16–17	Language and Communication
Municipal Capital (10,000 inhabitants)	Public	5º	24–26	Language and Communication
		6º	16–23	Music Education (National Holidays)
		6º	17–19	Class Council (Tutorships) and Natural Sciences
Regional Capital (230,000 inhabitants)	Subsidized (Coordinated)	5º	39–40	Class Council (Tutorships)
		5º	34–40	Class Council (Tutorships)
		6º	33–42	Class Council (Tutorships)
		6º	26–40	Class Council (Tutorships)

Source: authors’ elaboration.

**Table 2 ijerph-20-05595-t002:** Conditioned behavior associated with the clear expression of emotions (A11).

R − 3	R − 2	R − 1	R0	R + 1	R + 2	R + 3	R + 4
			**A11**				
			E3				

*Note*: Conditioned behaviors (category) in bold letters represent an activating relationship.

**Table 3 ijerph-20-05595-t003:** Conditioned behaviors associated with confused expression of emotions (A12).

R − 3	R − 2	R − 1	R0	R + 1	R + 2	R + 3	R + 4
	*E*1	E1	**A12**	E1	*E1*	E1	*E1*
		*E5*	E5	*E3*		*E3*	
			E4	*E5*			
			*E1*				

*Note*: Conditioned behaviors (categories) in bold letters represent an activating relationship and those in italics represent an inhibitory relationship.

**Table 4 ijerph-20-05595-t004:** Conditioned behaviors associated with the behavior of associating personal emotions to personal behaviors (A21).

R − 3	R − 2	R − 1	R0	R + 1	R + 2	R + 3	R + 4
			**A21**		S1		

*Note*: The conditioned behavior (category) in bold letters represents an activating relationship.

**Table 5 ijerph-20-05595-t005:** Conditioned behaviors associated with the behavior of showing awareness of other people’s emotional reactions (A31).

R − 3	R − 2	R − 1	R0	R + 1	R + 2	R + 3	R + 4
	E1		**A31**	E4	E1	E4	E1
			E1	C2142			
			*E4*	*E1*			

*Note*: Conditioned behaviors (categories) in bold letters represent an activating relationship and those in italics represent an inhibiting relationship.

**Table 6 ijerph-20-05595-t006:** Conditioned behaviors associated with the behavior of relating other people’s behaviors to their emotions (A41).

R − 3	R − 2	R − 1	R0	R + 1	R + 2	R + 3	R + 4
	E1	E3	**A41**	E3		E3	
		*E1*	E1				
			E2				
			*E5*				

*Note*: Conditioned behaviors (categories) in bold letters represent an activating relationship and those in italics represent an inhibitory relationship.

**Table 7 ijerph-20-05595-t007:** Multiple cases: clear expression of emotions (A11) as a focal behavior with category selection as conditioned behaviors.

Class	Quadrant 1	Quadrant 2	Quadrant 3	Quadrant 4
1	**A31**, **A41**, B14, D1, D24, D31			A51, B12, *D21*, D35
2	**A31**, C11, D35	C12	**S2**, C13, B24	S1, B21
3	A21, **A31**, **A41**, B12, B14, **C12**, S2, D33		B11, **C11**	D1, *D*21, D31
4		B11	**C11**	S1
5		S1		S2
6	**A31**, **C12**, S1, B24	B11	**C11**, D21, D31	B14
7	A12, **A41**, **C12**, C2142, S1, B24, D1, D24, D32, D35		C13, **S2**, B21	A51
8	**A31**, **A41**, **C12**,	D22	**S2**	
9	A12, **A31**, **A41**, B11, **C12**, B21, D1, D23, D24, D31, D33, D34, D35		**C11**, C2142, S1, **S2**	B24, *D21*

*Note*: Categories that are repeated in the same quadrant more than 3 times have been identified in italics and those categories present more than four times in the same quadrant have been highlighted in bold.

**Table 8 ijerph-20-05595-t008:** Multiple cases: confused expression of emotions (A12) as a focal behavior with category selection as conditioned behaviors.

CLASS	Quadrant 1	Quadrant 2	Quadrant 3	Quadrant 4
1	**A31**, **A41**, *B13*, **C12**	A21, C11	B23, D34	B24, C13, D1, D35
2	**A31**, **A41**, A51, **C12**, C2142, *D33*, *D34*, **D35**	A21, B14	B13, **S1**, **S2**	B12, C13, B22, D21
3	**A31**, **A41**, *B13*, **C12**, *C13*, *B23*, *B24*, **D1**, **D21**, **D24**, **D31**, *D32*, *D33*, *D34*, **D35**		**C11**	A51
4	**A41**, B11, B14, **C12**, *C13*, *B23*, **D1**, **D21**, D22; D23, **D24**, **D31**, *D32*, **D35**		**C11**, **S2**	S1
5	**A41**, B12, **C12**, S2; B22; **D1**, **D21**, **D24**, **D31**	D34	B11, **C11**, **S1**	
6	*B13*, **C12**, *C13*, S1, B22, *B24*, **D1**, *D32*, **D35**	A41, C2142	**C11**, **S2**, B21	A21
7	A11, **A31**, **A41**, **C12**, C2141, S1, *B24*, D23	B22, D1	C13, **S2**, B21, B23	
8	A21, **A41**, **C12**, S2, *B23*, **D1**, **D21**, **D24**, **D31**, **D35**		**C11**, **S1**, B22	B12, D23
9	A11, A21, **A31**, **A41**, B11, B14, **C12**, **D1**, **D24**, *D33*, *D34*, **D35**	S2	B13, C13, C2142, **S1**	D22

*Note*: Categories that are repeated in the same quadrant more than 3 times have been identified in italics and those categories present more than four times in the same quadrant have been highlighted in bold.

**Table 9 ijerph-20-05595-t009:** Multiple cases: associating personal behaviors to his/her own mood states (A21) as focal behavior with category selection as conditioned behaviors.

Class	Quadrant 1	Quadrant 2	Quadrant 3	Quadrant 4
1	A31, **A41**, *A51*, B14, *D1*, D32, E7, S1		**S2**	D35, A12
2	**C11**		**S2**	D1, D21, D31, D35, A12
3	A11, **A41**, *A51*, B13, B23, C12, *D1*, **D21**, D24, D31, D33,	E6	C11	
4				
5	**C11**	C2141, E5	**S2**	
6	**A41**, *A51*, *B11*, B12, **C11**, S1, **D21**, D24, D31, *D35*, E1,	D22, E5, A12	C13, **S2**	
7	S2, C2141, D1		B21, *S1*	D21, D31
8	A12, **A41**, *B11*, B23, C12, *D1*, **D21**, *D35*, E1, E3, E7,	A51	*S1*, E2, E5	
9	A12, A31, **A41**, **C11**, C12, *B11*, B21, **D21**, D23, D33, *D35*, E5	D1, D31, E4	C13, *S1*, **S2**, B13, B14, C2142	B24

*Note*: Categories that are repeated in the same quadrant more than 3 times have been identified in italics and those categories present more than four times in the same quadrant have been highlighted in bold.

**Table 10 ijerph-20-05595-t010:** Multiple cases: showing awareness of other people’s emotional reactions (A31) as a focal behavior with category selection as conditioned behaviors.

Class	Quadrant 1	Quadrant 2	Quadrant 3	Quadrant 4
1	**A11**, **A12**, A21, **A41**, **C12**, *D24*, C2142, D34, E7,	S1, D33	*B12*, B24, **C13**, **S2**, D21	
2	**A11**, **A12**, **B11**, **C11**, *D24*, *D35*, E7,		**C13**, **S2**, D21, E1	D1
3	**A11**, **A12**, **A41**, **B11**, **C12**,	S2	B14, S1	D24, D31, D35, E5, A51
4	**A41**	B13, B22		B23, D23
5	**A41**, **B11**, **C11**, C2141, E4,		**S2**, **E2**	B14, B21
6	**A11**, **A41**, B21, **C11**, E5, S1	D32, D35	*B12*, B13, B22, *B23*, B24, C12, **C13**, **S2**, **E2**	C2142, D1, D31
7	**A12**, **A41**, B24, **C11**. **C12**, *D1*, S1	D21, D22, D31, E4	*B23*, **C13**, **S2**,	E5
8	**A11**, **A41**, *D1*, D21, *D35*, E4,	S1, D22	**E2**	
9	**A11**, **A12**, A21, **A41**, **B11**, B21, **C12**, *D1*, D21, D23, *D24*, D31, D33, D34, *D35*, E1, E5	C11	*B12*, B22, *B23*, **C13**, D22, D32, **E2**, S1, **S2**	B24

*Note*: Categories that are repeated in the same quadrant more than 3 times have been identified in italics and those categories present more than four times in the same quadrant have been highlighted in bold.

**Table 11 ijerph-20-05595-t011:** Multiple cases: relating other people’s behaviors to their emotions (A41) as a focal behavior with category selection as conditioned behaviors.

Class	Quadrant 1	Quadrant 2	Quadrant 3	Quadrant 4
1	**A11**, **A12**, **A21**, **A31**, **C12**, **D1**, D24, C2142, D32, D33, D34, **E3**, E7		B12, **C13**, **S2**, *D21*, E1	B24
2	**A12**, **A51**, **B11**, **C11**, **D35**, **E3**, *E4*,	C2142	B12, B22, *B23*, B24, **C13**, S1, **S2**, D1, *D21*, D24, D31, E2	
3	**A11**, **A12**, **A21**, **A31**, **A51**, B22, **C12**, D33, **D35**,	E6	C11, **S2**	C2142, D21, D31
4	**A12**, **A31**, **C12**, **D1**, D23,		B14, C11, **S2**, E2, E6	D24
5	**A12**, **A31**, **B11**, B13, B21, **C11**, E1, *E4*,	C12, D35	B22, B24; **C13**, **S2**, D1, *D21*, D31, E2	E3
6	**A21**, **A31**, **A51**, **B11**, B22, **C11**, **D1**, D22, **D21**, D31, D32, **D35**, **E3**, E5,		*B23*, C12, D23, E2	A12
7	**A11**, **A12, A31**, **A51**, B14, **C12**, S1, **D1**, **D21**, D31, **D35**, **E3**,		*B23*, **C13**, **S2**	E5
8	**A11**, **A12**, **A21**, **A31**, **A51**, **B11**, B13, B23, **D1**, **D21**, D22, D24, **D35**, **E3**,	C12, D32, E7	S1, **S2**, E2	C11, E4
9	**A11**, **A12**, **A21**, **A31**, **C11**, **C12**, **B11**, **D21**, D23, **D35**, *E4*, E5	B23, D33	**C13**, S1, **S2**, B13, B14, C2141, C2142	B24

*Note*: Categories that are repeated in the same quadrant more than 3 times have been identified in italics and those categories present more than four times in the same quadrant have been highlighted in bold.

**Table 12 ijerph-20-05595-t012:** Multiple cases: differentiating between personal and other people’s emotions (A51) as a focal B = behavior with category selection as conditioned behaviors.

Class	Quadrant 1	Quadrant 2	Quadrant 3	Quadrant 4
1	B22, *A21*	S2, D32, A11	D21	
2	B24, S2, E3, A12, **A41**	C2141		
3	B23, *A21*, **A41**	B13, B14, C12, D1, E5, A12, A31		B22, C11
4				
5				
6	B21, C11, S1, D21, D31, *A21*, **A41**	C2142	B23, C12, C13, *S2*	
7	B14, B24, C12, S1, D24, D35, **A41**	D32, A11	B11, C13, *S2*, E1	B12, B13, C11, D31
8	B11, B23, C11, D1, E3, **A41**	E7	S1, *S2*	D35, A21
9				

*Note*: Categories that are repeated in the same quadrant more than 3 times have been identified in italics and those categories present more than four times in the same quadrant have been highlighted in bold.

## Data Availability

Data can be shared privately by writing to marina.alarcon@ufrontera.cl.

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
