# Peer review of "Systematic Observation of Emotional Regulation in the School Classroom: A Contribution to the Mental Health of New Generations"

_ijerph, 2023, doi:10.3390/ijerph20085595_

Round 1

Reviewer 1 Report

 The current paper aimed of analyzing the forms of expression and regulation of emotions that are observed in the school classroom, through a mixed method study that systematically observes 9 classes during 5 sessions each.

Overall, the paper is well-written, and recent literature on the regulation of emotions is properly quoted in the introduction.

The research is very interesting and of definite relevance to the field of studies on emotional regulation within schools. In general, the bibliography and reference literature is well presented. The results are presented comprehensively and with a certain scientific approach. My doubt is on the presentation of the methodology and results, not on the analysis of the data but just on the way the authors decided to present the methodological part. Of course, it is obvious that scientific journals are not popular but are for scholars. But the research gives known some methodological content that in my opinion should instead necessarily be described even in a scientific article like this one.

Lines 31-34 “Emotions help people adapt to the opportunities and challenges they face; they involve the combination of innate, lived and learned factors, constituting a set of bodily, cognitive, and behavioral patterns and responses with which individuals approach complex perceptual, bodily, and motivational processes when interacting with other people”. I think it is an 'irrelevant introduction. The article can start directly from line 36.

Lines 77-80: “It is necessary to mention that, traditionally, schools have not considered emotional education, focusing on conveying theoretical knowledge and attending to emotional issues basically when dealing with affective and/or behavioral problems in the diversity of the school classroom”. I think it is a very strong generalization as well as simplistic.

Line 94: “the quantitizing  proposal of Anguera et al”. Please try to describe a little bit deeper.

Lines 115-117: “The ad-hoc designed instrument”. Please try to define it a little bit deeper.

Lines 11-117: Dimensions and subdimensions, configuring category systems for each subdimension called "Guideline for the Observation of Communication and Emotional Self-regulation". These dimensions should be presented and explained before the results section, otherwise, it is very confusing.

Lines 120: Why just 15% of the data were coded at three different times?

Line 182:  “each category considered as behavioral criterion”. Again turns out to be unclear.

Author Response

We are pleased to respond to the observations made about our manuscript, Systematic Observation of Emotional Regulation in the School Classroom: A Contribution to the Mental Health of the New Generations.

Revisor 1

  • Lines 31-34 “Emotions help people adapt to the opportunities and challenges they face; they involve the combination of innate, lived and learned factors, constituting a set of bodily, cognitive, and behavioral patterns and responses with which individuals approach complex perceptual, bodily, and motivational processes when interacting with other people”. I think it is an 'irrelevant introduction. The article can start directly from line 36.

This text has been deleted.

  • Lines 77-80: “It is necessary to mention that, traditionally, schools have not considered emotional education, focusing on conveying theoretical knowledge and attending to emotional issues basically when dealing with affective and/or behavioral problems in the diversity of the school classroom”. I think it is a very strong generalization as well as simplistic.

The reference that points to the statement has been placed explicitly and the concept “traditionally” has been eliminated.

In this regard, it is necessary to mention that, according to García-Lázaro et al. [25], schools have not considered emotional education, focusing on conveying theoretical knowledge and attending to emotional issues basically when dealing with affective and/or behavioral problems in the diversity of the school classroom

  • Line 94: “the quantitizing proposal of Anguera et al”. Please try to describe a little bit deeper.

This approach can transform the information differently to how methods mixed methods are described in the literature, and consists of the succession of three stages QUAL-QUAN-QUAL: In the first QUAL stage, descriptive records are obtained, and from both this and the theoretical framework, a defined structure – observation instrument – is built to systematize these in the form of a code matrix. In the QUAN stage, the code matrices are analyzed quantitatively, and in the second QUAL stage the results are interpreted, returning to the initial problem.

  • Lines 115-117: “The ad-hoc designed instrument”. Please try to define it a little bit deeper.
  • Lines 11-117: Dimensions and subdimensions, configuring category systems for each subdimension called "Guideline for the Observation of Communication and Emotional Self-regulation". These dimensions should be presented and explained before the results section, otherwise, it is very confusing

The explanation of the instrument has been strengthened, as explained in the following:

The coding was performed using the ad-hoc designed instrument, that combines field format with category systems, [32] and the structure of which has been prepared from the descriptive records and from the theoretical framework, gradually generating conceptual macrodimensions deployed in dimensions and subdimensions, and from each of which a category system (exhaustive and mutually exclusive) or behavior catalog (mutually exclusive) has been constructed. This observation instrument, which has been structured from dimensions and subdimensions, configuring category systems for each subdimension called "Guideline for the Observation of Communication and Emotional Self-regulation" (OCAE) [27] is formed by the four following macrodimensions: (A) knowledge of one’s own emotions and those of others, (B) self-regulation of behavior for the objective to be achieved, (C) ability to establish assertive communication, and (D) an effective approach to conflict. Each of these was introduced at several levels, and the respective category systems and behavior catalogs were prepared until progressively obtaining the definitive observation instrument [33].

  • Lines 120: Why just 15% of the data were coded at three different times?

The paragraph has been modified to read as follows:

In order to guarantee the quality of the data obtained, and considering that in the scientific literature in different areas it is common to use a percentage of records included between 5 and 20% to find agreement (Lapresa et al. [34], 15% of the data was recorded. To guarantee the greatest rigor in the quality control of the data, the records were coded at three different times, and Cohen's kappa coefficient was calculated [28], obtained by computer using the free program GSEQ5, with multi-event data, type II [29], achieving a concordance of 0.60 or greater.

  • Line 182:  “each category considered as behavioral criterion”. Again turns out to be unclear.

We have observed that the English name for “conducta criterio” is “given behavior”, and have therefore corrected the concept where necessary.

Reviewer 2 Report

    Dear Editors and Authors,   I am writing to review the article, “Systematic Observation of Emotional Regulation in the School Classroom: A Contribution to the Mental Health of the New Generations”. Overall, the article provides a strong overview of the importance of emotional development and regulation in schoolchildren. The authors have done a good job of providing a comprehensive overview of the topic and citing relevant research.   However, the article could be improved with a few minor revisions. For example, the introduction could be improved by providing more detail on the specific strategies that can be used to foster emotional development and regulation in children in existing literature. Additionally, the article could benefit from providing more examples of how the school environment can facilitate emotional development and regulation. This would help to understand the current implications of your study.   The conclusion of the article is well written and provides a good overview of the research and its findings. However, there could be some improvement in terms of providing more clarity on the implications of the research and how the findings can be applied in practice. Additionally, the authors could provide more detail on the statements of Gómez & Calleja, Garaigordóbil, and other relevant sources, to better explain the importance of their findings and how they can be used in the classroom.   I hope that these suggestions are helpful. I look forward to reading the revised version of the article.   Sincerely, Reviewer

Author Response

We are pleased to respond to the observations made about our manuscript, Systematic Observation of Emotional Regulation in the School Classroom: A Contribution to the Mental Health of the New Generations.

Revisor 2

  • The introduction could be improved by providing more detail on the specific strategies that can be used to foster emotional development and regulation in children in existing literature.

We accept the suggestion and have added the following text:

It has been observed that the social support on offer within the school affects the students’ perception of subjective well-being, underscoring the importance of accepting and being aware of the emotional responses in the construction of social relationships in the school, which involves strategies linked to feelings of being accepted, respected, included and supported by others [9].

  • Additionally, the article could benefit from providing more examples of how the school environment can facilitate emotional development and regulation. This would help to understand the current implications of your study.

We accept the suggestion and have added the following text:

When students have more adult support in school, they exhibit more positive emotions during learning activities as well as greater effort, focus, and persistence by initiating and taking part in learning activities. In this regard, Frivold Kostøl & Cameron [10] have noted that in order to mitigate the possible adverse effects of emotion, teachers could develop behaviors that support effective emotional regulation: (a) help the students control their behavior when they are angry, (b) facilitate emotional awareness in students and the acceptance of emotions through participation in conversations and debates with the students on the range of emotional responses in the classroom and life events, and (c) allow the expression of emotions that the students can experience, supporting them to express their thoughts and feelings.

According to Riquelme [11] and Riquelme et al. [12], this can be done through strategies such as theater, music, or critical reading; or favor peer dialogue processes as a way to internalize important knowledge [13].

  • There could be some improvement in terms of providing more clarity on the implications of the research and how the findings can be applied in practice.

We accept the suggestion and have added the following text:

In connection with the above, Martínez-Maldonado et al. [46] noted a strong weight of the school context and student characteristics in the teaching practices observed and in the classroom interactions, which primarily present high direction of the teacher towards the students, absence of student-student interaction and high student-knowledge interaction time.

  • Additionally, the authors could provide more detail on the statements of Gómez & Calleja, Garaigordóbil, and other relevant sources, to better explain the importance of their findings and how they can be used in the classroom.

We accept the suggestion and have added the following text:

In this sense, the suggestions by Ferreira et al. [51] become relevant, insofar as socioemotional learning must be based on a systemic perspective, where the development of socioemotional skills occurs not only in the classroom but also in the entire school, in families, and in the school community. This will fully contribute to the education process since, as Berg et al. [52] and Greenberg et al. [53] indicate, the relation between school climate and socioemotional competences is bidirectional, i.e., in the same way that school climate affects or influences socioemotional competences, socioemotional competences affect or influence school climate.

  • In making the requested changes, we needed to add new references and thus modify both the references section as well as their numeration in the text. The references are as follows:

References

  1. Saarni, C. (1999). The development of emotional competence. Guilford Press.
  2. Montoya, I., Postigo, S. & Villena, L. (2018). La regulación de las emociones. En M. Giménez – Dasí. & L. Quintanilla Cobián (Coords.), Desarrollo emocional en los primeros años de vida. (pp. 71-82). Pirámide.
  3. Rieffe, C. (2016). La regulación emocional infantil en el contexto social. En R. González Barrón y L. Villanueva Badenes (Coords.), Recursos para educar en emociones. De la teoría a la acción. (pp. 125-149). Pirámide.
  4. Gómez, O., & Calleja, N., (2016). Regulación emocional: definición, red nomológica y medición. Revista Mexicana de Investigación en Psicología, 8(1) 96-117.
  5. Segundo, J.; Cantos, A.L.; Ontiveros, G.; O’Leary, K.D. (2022) Risk Factors of Female-Perpetrated Intimate Partner Violence among Hispanic Young Adults: Attachment Style, Emotional Dysregulation, and Negative Childhood Experiences. J. Environ. Res. Public Health, 19, 13850. https://doi.org/10.3390/ijerph192113850
  6. Ordoñez, A. & González, R. (2016). Las emociones y constructos afines. En R. González Barrón & L. Villanueva Badenes (Coords.), Recursos para educar en emociones. De la teoría a la acción (pp. 21-54). Pirámide.
  7. Villanueva, L. & Górriz, A. B. (2016). La conciencia emocional y su relación con el bienestar infantil y juvenil. En R. González Barrón & L. Villanueva Badenes (Coords.), Recursos para educar en emociones. De la teoría a la acción. (pp.151-172). Pirámide.
  8. Marchesi, A., (2017). Alumnos con dificultades sociales, emocionales y de conducta. En A. Marchesi, C. Coll, & J. Palacios (Compiladores), Desarrollo Psicológico y Educación. 3. Respuestas educativas a las dificultades de aprendizaje y del desarrollo. 3ª Ed. (pp. 255-285). Alianza.
  9. Jing Li & Meilin Yao & Hongrui Liu, 2021. "From Social Support to Adolescents’ Subjective Well-Being: the Mediating Role of Emotion Regulation and Prosocial Behavior and Gender Difference," Child Indicators Research, Springer;The International Society of Child Indicators (ISCI), vol. 14(1), pages 77-93
  10. Frivold Kostøl, E. M., & Cameron, D. L. (2021). Teachers’ responses to children in emotional distress: A study of co-regulation in the first year of primary school in Norway. Education 3-13, 49(7),821–831. https://doi.org/10.1080/03004279.2020.1800062.
  11. Riquelme, E. (2013). La lectura mediada de literatura infantil como herramienta para el desarrollo de competencias emocionales. Tesis doctoral. Universidad Autónoma de Madrid.
  12. Riquelme, E., Munita, F., Jara, E., y Montero, I. (2013). Reconocimiento facial de emociones y desarrollo de la empatía mediante la lectura mediada de literatura infantil. Cultura y Educación, 25(3), 375-388.
  13. Giménez-Dasí, M.; Quintanilla, L.; Daniel, M.-F. (2013). Improving Emotion Comprehension and Social Skills in Early Childhood through Philosophy for Children. Childhood & Philosophy, 9(17), 63-89
  14. DÅ‚ugosz, P.; Liszka, D. & Yuzva, L. (2022). The Link between Subjective Religiosity, Social Support, and Mental Health among Young Students in Eastern Europe during the COVID-19 Pandemic: A Cross-Sectional Study of Poland and Ukraine. J. Environ. Res. Public Health, 19, 6446. https://doi.org/10.3390/ijerph19116446
  15. Hua, Z. & Ma, D. (2022). Depression and Perceived Social Support among Unemployed Youths in China: Investigating the Roles of Emotion-Regulation Difficulties and Self-Efficacy. J. Environ. Res. Public Health, 19, 4676. https://doi.org/10.3390/ijerph19084676
  16. Bisquerra, R. (Coord.); Punset, E., Mora, F., García Navarro, E., López-Cassà, È., Pérez-González, J. C., Lantieri, L., Nambiar, M., Aguilera, P., Segovia, N. & Planells, O. (2012). ¿Cómo educar las emociones? La inteligencia emocional en la infancia y la adolescencia. Esplugues de Llobregat (Barcelona): Hospital Sant Joan de Déu. Editoral Faros.
  17. Burguet, M. & Buxarrais, M. R. (2013). La Eticidad de las TIC. Las Competencias Transversales y Sus Paradojas. Tesi14 (3), 87-100.
  18. Cabero, J., & Romero, R. (2001). Violencia, juventud y medios de comunicación. Comunicar, 17, 126-132.
  19. Ramirez, L. (2018). Desarrollo sociomoral y educación para la paz: construyendo entornos favorables para el desarrollo de competencias para la ciudadanía. Avances en Psicología Latinoamericana, 36(2), 227-233. https://doi.org/10.12804/revistas.urosario.edu.co/apl/a.6748
  20. Bisquerra, R. (2003). Educación emocional y competencias básicas para la vida. Revista de Investigación Educativa, 21(1), 7-43.
  21. Buxarrais, M. R. (2013). Nuevos Valores para una Nueva Sociedad. Un Cambio de Paradigma en Educación. Edetania, 43, 53-65.
  22. Moro, M. (2007). Educación en valores a través de la publicidad de televisión. Comunicar, 28, 183-189.
  23. Prado, J. (2001). La competencia comunicativa en el entorno tecnológico: desafío para la enseñanza. Comunicar, 17, 21-30.
  24. Rodríguez-Gómez, D., Castro, D., & Meneses, J. (2018). Usos problemáticos de las TIC entre jóvenes en su vida personal y escolar. Comunicar, 56, 91-100. https://doi.org/10.3916/C56-2018-09
  25. García-Lázaro, I., Gallardo-López, J. A., & López-Noguero, F. (2019). La inteligencia emocional y la educación emocional en la escuela: un estado de la cuestión a través del análisis bibliométrico de la producción científica en Scopus (2015- 2019). En J. A. Marín Marín, G. Gómez García, M. Ramos Navas-Parejo & N. Campos Soto (Eds.), Inclusión, Tecnología y Sociedad: investigación e innovación en educación (pp. 220-231).
  26. Izquierdo, C. (2000). Comunicación interpersonal y crecimiento emocional en centros educativos: un modelo interpretativo. Educar, 26, 127 -149.
  27. Anguera, M. T. (2010). Posibilidades y relevancia de la observación sistemática por el profesional de la Psicología. Papeles del Psicólogo, 31(1), 122-130. https://bit.ly/3JDv8zA
  28. Anguera, M. T., Portell, P., Hernández-Mendo, A., Sánchez-Algarra, P., & Jonsson, G. K. (2021). Diachronic analysis of qualitative data. In A. J. Onwuegbuzie & B. Johnson (Eds.), Reviewer’s Guide for Mixed Methods Research Analysis (pp. 125-138). Routledge.
  29. Creswell, J. W. & Plano Clark, V. L. (2011). Designing and Conducting Mixed Methods Research, 3rd ed. Sage.
  30. Anguera, M. T., Blanco-Villaseñor, A., Losada J. L., & Sánchez-Algarra, P. (2020). Integración de elementos cualitativos y cuantitativos en metodología observacional. Ámbitos. Revista Internacional de Comunicación 49, pp. 49-70.
http//doi.org/10.12795/Ambitos.2020.i49.04
  31. Anguera, M. T., Blanco-Villaseñor, A., & Losada, J. L. (2001). Diseños observacionales, cuestión clave en el proceso de la metodología observacional. Metodología de las Ciencias del Comportamiento, 3(2), 135-161.
  32. Anguera, M.T., Blanco-Villaseñor, A., Losada, J.L., y Portell, M. (2018). Pautas para elaborar trabajos que utilizan la metodología observacional. Anuario de Psicología, 48, 9-17. https://doi.org/10.1016/j.anpsic.2018.02.001
  33. Alarcón-Espinoza, M. (2021). Autorregulación Emocional en la Cotidianeidad de la Vida Escolar: Observación Sistemática en Aulas con Estudiantes Chilenos de 10 a 12 años. [Tesis Doctoral, Universidad de Barcelona]. Repositorio Institucional - Universidad de Barcelona.
  34. Lapresa, D., Otero, A., Arana, J., Álvarez, I., y Anguera, M.T. (2021). Concordancia consensuada en metodología observacional: efectos del tamaño del grupo en el tiempo y la calidad del registro. Cuadernos de Psicología del Deporte, 21(2), 47-58. https://doi.org/10.6018/cpd.467701
  35. Cohen, J. (1968). Weighted kappa: Nominal scale agreement with provision for scaled disagreement of partial credit. Psychological Bulletin, 70, 213-220.
  36. Bakeman, R. (1978). Untangling streams of behavior: Sequential analysis of observation data. In G. P. Sackett (Ed.), Observing Behavior, Vol 2: Data collection and analysis methods (pp. 63-78). University of Park Press.
  37. Anguera, M. T. & Hernández-Mendo, A. (2015). Técnicas de análisis en estudios observacionales en ciencias del deporte. Cuadernos de Psicología del Deporte, 15(1), 13-30. https://dx.doi.org/10.4321/S1578-84232015000100002
  38. Bakeman, R. & Quera, V. (2011). Sequential analysis ad observational methods for the behavioral sciences. Cambridge University Press.
  39. Sackett, G. P. (1980) Lag sequential analysis as a data reduction technique in social interaction research. In D. B. Sawin, R. C. Hawkins, L. O. Walker & J. H. Penticuff (Eds.), Exceptional infant. Psychosocial risks in infant-environment transactions (pp. 300-340). Brunner/Mazel.
  40. Cochran, W. G. (1954). Some methods for strengthening the common χ2 tests. Biometrics 10, 417-451. http//doi.org/10.2307/3001616.
  41. Hernández-Mendo, A., López, J. A., Castellano, J., Morales, V., & Pastrana, J. L. (2012). HOISAN 1.2: Programa informático para uso en Metodología Observacional. Cuadernos de Psicología del Deporte, 2(1), 55-77.
  42. Anguera, M. T. (2018). Del caso único al caso múltiple en el estudio del comportamiento humano. En Academia de Psicología de España, Psicología para un mundo sostenible, vol, II (pp.31-50). Pirámide. [I.S.B.N. 978-84-368-3889-3]
  43. Stake, R. E. (2006). Multiple case study analysis. Guilford Press.
  44. Yin, R. K. (2014). Case study research. Design and methods, 5th edition. Sage.
  45. Encinas, M. (2018). El papel de las emociones en el aula: Una aproximación histórico-cultural. En M. Giménez – Dasí, & L. Quintanilla Cobián (Coords.), Desarrollo emocional en los primeros años de vida (177-190). Pirámide.
  46. Martínez-Maldonado,P.; Armengol Asparó, C.; Muñoz Moreno, J. L. (2019).Interacciones en el aula desde prácticas pedagógicas efectivas Revista de Estudios y Experiencias en Educación, vol. 18, núm. 36, https://doi.org/10.21703/rexe.20191836martinez13
  47. García Amilburu, M., & García Gutiérrez, J., (2017). Conceptualización y ámbitos del proceso educativo. En Filosofia de la Educación, cuestiones de hoy y siempre (pp. 47-62). Narcea.
  48. Marchesi, A., (2017). Enfoques y estrategias educativas para personalizar la enseñanza. En A. Marchesi, C. Coll, & J. Palacios (Compiladores), Desarrollo Psicológico y Educación. 3. Respuestas educativas a las dificultades de aprendizaje y del desarrollo. 3ª Ed. (pp.83-109). Alianza.
  49. Lucas-Mangas, S.; Valdivieso-León, L.; Espinoza-Díaz, I.M. & Tous-Pallarés, J. (2022). Emotional
    Intelligence, Psychological Well-Being and Burnout of Active and In-Training Teachers. J. Environ. Res. Public Health, 19, 3514. https://doi.org/10.3390/ ijerph1906351
  50. Garaigordóbil, M. (2014). Conducta prosocial: el papel de la cultura, la familia, la escuela y la personalidad. Revista Mexicana de Investigación en Psicología, 6(2), 146-157.
  51. Ferreira, M., Reis-Jorge, J., Olcina-Sempere, G. y Fernandes, R. (2023). El aprendizaje socioemocional en la Educación Primaria: una investigación sobre las concepciones y las prácticas de los maestros en el aula. Revista Colombiana de Educación, (87), 37-60. https://doi.org/10.17227/rce.num87-12704
  52. Berg, J., Osher, D., Moroney, D. y Yoder, N. (2017). The intersection of school climate and social and emotional development. American Institutes for Research. https://www.air.org/sites/default/files/downloads/report/IntersectionSchool-Climate-and-Social-and-Emotional-Development-
  53. Greenberg, M., Domitrovich, C., Weissberg, R. y Durlak, J. (2017). Social and emotional learning as a public health approach to education. The Future of Children, 27(1), 13-32. https://doi.org/10.1353/foc.2017.0001

  • In addition, we modified the access link to the GSEQ V program, adding the correct one, which is:

https://www.mangold-international.com/en/products/software/gseq.html

  • In addition to the above, we added acknowledgements that we forgot on the first occasion and which we hope can be included. These are:

The first and third authors are grateful for the support of the Generalitat de Catalunya Research Group, GRUP DE RECERCA I INNOVACIÓ EN DISSENYS (GRID). Tecnología i aplicació multimedia i digital als dissenys observacionals [Grant number 2021 SGR 00718] (2022-2024).

The third author gratefully acknowledges the support of Spanish government project Integración entre datos observacionales y datos provenientes de sensores externos: Evolución del software LINCE PLUS y desarrollo de la aplicación móvil para la optimización del deporte y la actividad física beneficiosa para la salud [EXP_74847] (2023). Ministry of Culture and Sport, National Sports Council, and the European Union.

We hope we have been able to respond to your requests and/or suggestions.

Sincerely,

Marina Alarcón Espinoza

Round 2

Reviewer 1 Report

In my opinion, the authors have responded to my main requests, and if the other reviewers agree with my view, the manuscript may now be suitable for publication